# Dual STDP processes at Purkinje cells contribute to distinct improvements in accuracy and speed of saccadic eye movements

Lorenzo Fruzzetti[1,2☯], Hari Teja Kalidindi[3,4☯]*, Alberto Antonietti[5], Cristiano Alessandro[6,7], Alice Geminiani[6], Claudia Casellato[6], Egidio Falotico[1,2‡]*, Egidio D'Angelo[6,8‡]

1 The BioRobotics Institute, Scuola Superiore Sant'Anna, Pontedera (Pisa), Italy, 2 Department of Excellence in Robotics and AI, Scuola Superiore Sant'Anna, Pisa, Italy, 3 Institute of Information and Communication Technologies, Electronics and Applied Mathematics, Universite Catholique de Louvain, Ottignies-Louvain-la-Neuve, Belgium, 4 Institute of Neuroscience, Universite Catholique de Louvain, Ottignies-Louvain-la-Neuve, Belgium, 5 Department of Electronics, Information and Bioengineering, Politecnico di Milano, Milano, Italy, 6 Department of Brain and Behavioral Sciences, University of Pavia, Italy, 7 School of Medicine and Surgery/Sport and Exercise Medicine, University of Milano-Bicocca, Milan, Italy, 8 IRCCS Mondino Foundation, Pavia, Italy

☯ These authors contributed equally to this work.
‡ EF and EDA also contributed equally to this work.
* hari.kalidindi@uclouvain.be (HK); egidio.falotico@santannapisa.it (EF)

**Data Availability Statement:** Datasets from the simulations, together with sample analysis and plotting code associated with this work are

## Abstract

Saccadic eye-movements play a crucial role in visuo-motor control by allowing rapid foveation onto new targets. However, the neural processes governing saccades adaptation are not fully understood. Saccades, due to the short-time of execution (20–100 ms) and the absence of sensory information for online feedback control, must be controlled in a ballistic manner. Incomplete measurements of the movement trajectory, such as the visual endpoint error, are supposedly used to form internal predictions about the movement kinematics resulting in predictive control. In order to characterize the synaptic and neural circuit mechanisms underlying predictive saccadic control, we have reconstructed the saccadic system in a digital controller embedding a spiking neural network of the cerebellum with spike timing-dependent plasticity (STDP) rules driving parallel fiber—Purkinje cell long-term potentiation and depression (LTP and LTD). This model implements a control policy based on a dual plasticity mechanism, resulting in the identification of the roles of LTP and LTD in regulating the overall quality of saccade kinematics: it turns out that LTD increases the accuracy by decreasing visual error and LTP increases the peak speed. The control policy also required cerebellar PCs to be divided into two subpopulations, characterized by burst or pause responses. To our knowledge, this is the first model that explains in mechanistic terms the visual error and peak speed regulation of ballistic eye movements in forward mode exploiting spike-timing to regulate firing in different populations of the neuronal network. This elementary model of saccades could be extended and applied to other more complex cases in which single jerks are concatenated to compose articulated and coordinated movements.

available on Zenodo doi.org/10.5281/zenodo.7113780.

**Funding:** This research has received funding from the European Union's Horizon 2020 Framework Programme for Research and Innovation under the Specific Grant No. 945539 (Human Brain Project SGA3). The funders had no role in study design, data collection and analysis, decision to publish, or preparation of the manuscript.

**Competing interests:** The authors have declared that no competing interests exist.

## Author summary

It has been suggested that the cerebellum plays a crucial role in oculomotor adaptation. Computationally, the cerebellum is described as a supervised learner whose activity can be adjusted by synaptic changes proportional to the amount of mismatch between expected and actual movements outcomes (sensorimotor-errors). However, the spike-timing-dependent-plasticity (STDP) underlying adaptation has been so far modeled in behaviors where the error is continuously available as tracking-error. Such models depend on detailed tracking-errors to improve movement quality. We asked if the cerebellum can maintain good motor control even if the error is not completely available? This is important considering that error-dependent-STDP is only a subset among a family of STDP processes in the cerebellum. Moreover, even the physiological signals that were generally thought to carry sensorimotor-errors (called complex-spikes) are shown to encode multiple types of information regarding the movement. In this context, we characterize the role of cerebellar STDPs in saccade control, where the error information is constrained because of sensory suppression. We show that even in the absence of detailed error, the cerebellum can leverage two of its STDPs to increase movement quality. Hence, we emphasize the need to go beyond error-centric view to understand how the cerebellum improves motor behavior.

## Introduction

Motor control in biological systems is commonly considered to depend on two interacting components, namely sensory feedback and prediction about the state of the body [1, 2]. Suppose we want to move our body from one *state* to another different state in a goal directed manner. One way to solve this goal-directed movement control problem is to find a continuous mapping from sensory feedback onto a set of muscle activations, referred to as a *control policy*, such that the body can be initialized at any state and reach the desired state. However, state feedback in biological motor systems is typically noisy, delayed due to information processing latency, or absent. Particularly, a class of eye movements called saccades occur in a purely ballistic manner, where the sensory feedback is completely suppressed from being useful for online motor control [3].

 Compensating for the delay or the lack of sensory feedback involves a putative state estimation process that uses internal prediction about the state of the body [1, 4]. To facilitate internal prediction, the motor system needs to keep track of the errors between predicted body movement and the actual movement, as perceived by the sensory outcomes [5]. This means that the nervous system should still be able to store and use state feedback regarding the actual movement for predictive control, even if it is not possible to directly use this sensory information for online feedback control. While a detailed movement information may be available in several movement behaviors [6], albeit with delay and noise, saccades occur at fast time-scales (20–100 ms) that result in the absence of such a precise temporal information. Neural correlates have been found in different regions, such as the superior colliculus (SC) and the cerebellum, that encode the end visual error (called foveation error) of saccadic eye movements [7–9]. Indeed, a neural signal that encodes the true error trajectory throughout the movement is not possible in the saccades because of *sensory suppression* during the movement [3]. Given the importance of sensory feedback and of a detailed error information for stable motor control,

we ask how does the nervous system compensate for the lack of both sensory feedback and a precise temporal error information in saccade behavior?

Saccadic eye-movements play a crucial role in visuo-motor control by allowing rapid foveation onto new targets [10, 11]. Several experimental reports indicate that the cerebellum is involved in predicting and controlling different kinematic aspects of saccades such as the eye movement amplitude, peak speed and duration [12–14]. Pharmacological inactivation of the cerebellar output has been found to produce erroneous saccades that overshoot the target, i.e., incur larger displacement than necessary, and simultaneously increase movement duration and reduce the peak eye speed. Notably, the absence of cerebellar contribution does not eliminate the ability to make saccades, but significantly affects the peak speed and accuracy of the movements [14, 15]. Hence, the control system can be comprised of an imprecise base controller connected with the cerebellum, which can act as an adaptive controller that fine tunes the sluggish base control policy [16, 17]. Several computational models focused on the cerebellum involvement in saccadic control [16, 18–20] and adaptation [21, 22] and accounted for residual behavioral response following cerebellar lesions [23, 24]. However, along with the absence of cellular recordings, these early models did not simulate how different spiking neural populations and different plasticity processes within the cerebellar circuitry provide computations supporting movement control and adaptation. The model proposed by [22] predicted the role of different types of neuronal spiking in the cerebellum in saccade control and adaptation, but did not explore the plasticity processes that might jointly influence peak speed and visual error of saccades. Studying if and how the cerebellar plasticity can jointly influence the peak speed and error is important, as recent behavioral experiments indicate that during saccade adaptation, the early and late components within a given movement are independently adjusted at different timescales [25, 26]. Such independent changes in early and late period motor commands can cause distinct changes in peak speed and visual error. However, the neural correlates of independent trial-by-trial adjustments of motor commands within a single movement are unknown. Notably, the activity in superior colliculus that is generally considered to convey a movement plan as input to the cerebellum [19], is shown to be unaltered across saccade adaptation trials [27]. In principle, the cerebellum can leverage its adaptive filter property [28–31], where multi-timescale changes in synaptic weights are used to alter the cerebellar output based on measurements of movement quality, while its input pattern is unaltered.

Recent experiments in [32, 33] provide multiple observations regarding how cerebellum encodes saccades, and how such representations are continuously reshaped on a trial-by-trial basis by error-driven adjustments of cerebellar neuronal activity. In [32], the authors recorded spiking activity from cerebellar Purkinje cell (PC) populations to investigate how the PC activity is related to the eye movements. Monkeys were trained to make repeated saccades towards radial targets displayed on a vertical screen. As the monkeys moved their eyes towards the targets, they recorded two different kinds of spikes emitted by PCs: high frequency Simple spikes (SSpikes) resulting from granule cell projections (GrC) onto PCs, and low frequency Complex spikes (CSpikes) resulting from Inferior Olive (IO) projections. In [32], it was observed that the SSpike activity of PC populations is correlated with the eye speed and displacement, and predicts the speed of the upcoming movement. Notably, the PCs that display similarity in their CSpike tuning to the direction of movement error have been grouped together in separate populations. In [33, 34], it was shown that CSpikes in PC populations carry information about visual foveation errors after the end of a given eye movement and modulate PC SSpikes on a trial-by-trial manner, to reduce errors incurred in subsequent movements. An important conceptual gap remains: if and how the CSpike-driven modulation of SSpikes in PC population helps in independently regulating both speed and errors of the eye movements. This is because

CSpikes were observed to be a function only of the end foveal error that does not carry information about movement speed [34].

A fundamental aspect that must be considered to correlate biological properties to behavior, is the way brain circuits encode information. In control theory, a common assumption is that a rate code might suffice. For example, recent models used a continuous rate-based representation of neural activity [17, 35] to reproduce the PC population responses during saccades to different visual targets. However, it has been proposed that cerebellar cortex relies on millisecond spike precision of PCs, not on individual firing rates, to convey to the nucleus when to stop a movement [36]. This implies that spike timing is critical and that spike-based modeling strategies are needed to face the issue. We therefore propose a spiking neural network emulating the cerebellar circuit that makes value of the information carried by the precise timing of spikes in the induction mechanisms of spike-timing-dependent plasticity (STDP) [28].

## Contribution

Following the predictions of our previous model [35], we hypothesize that the cerebellum can regulate saccade kinematics by two distinct STDP processes that individually influence the end foveal error and peak speed. We know from previous studies that the strength of synapses from parallel fibers (PF) to PCs is subject to changes by multiple STDP mechanisms such as long-term depression (LTD) and long-term potentiation (LTP) [37–40]. Through a realistic spiking network model of cerebellum [41–44] with a novel STDP recombination rule, we show that the errors incurred towards a given visual target can be rapidly decreased by means of an LTD process at PF-PC synapses, driven by CSpikes that indicate the end foveal error, occurring ~100 ms after the end of each eye movement. On the other hand, movement peak speed is shown to be increased by an LTP process in PF-PC synapses that occurs due to a reduced probability or even absence of CSpikes during the movement, and can increase the SSpikes during the eye movement on a trial-by-trial basis. Importantly, only when simulating such a dual plasticity process, PC populations increase their activity on a trial-by-trial basis in the anticipatory and early movement period, leading to both accurate and faster eye movements, similar to the experimental observations in [32].

Cerebellar PCs *in vivo* show a combination of burst and pause patterns with respect to their spontaneous activity levels, depending upon whether a given PC receives dominant excitatory inputs from granule cells or inhibitory inputs from molecular layer interneurons, forming effective "bursters and pausers" PC subpopulations [45]. Our model produces the testable prediction that adaptive control of saccades involves LTP and LTD in specific burst and pause PC subpopulations projecting onto a common deep cerebellar nucleus thereby. Ablating plasticity at either burst or pause subpopulations prevents the circuit from increasing movement peak speed, while an intact plasticity at any one of these subpopulations maintains the ability to reduce movement errors (or to improve movement accuracy). In summary, the present model correlates neural and behavioral variables, anticipating the synaptic mechanisms that would lead to control of accuracy and speed of saccades.

## Results

### Dual STDP plasticity in PC populations decreases foveal error and increases peak speed

In this work, a given saccade is considered to be accurate if it incurs a low error, defined as the difference between the target angular displacement and the end movement displacement. We used errors expressed in *degrees* of rotation, but the same model can work with retinotopic or

pixel difference between target and the end foveation locations [35]. When the end movement displacement is higher than the target displacement i.e., overshooting movement, it incurs a positive error, while an undershooting movement incurs a negative error.

There are two main ways by which the given saccade controller can modulate the saccades. First, STDP mechanisms influence the strength of PF-PC synapses such that the PC population activity is modified. In this case the total input MF drive to the cerebellum from the superior colliculus can remain unchanged across trials. Second, the modulation of neural activity occurs upstream of the cerebellum (e.g., in the superior colliculus), and influences the cerebellum to produce a different PC output. Here, we consider the former process focusing on the effect of STDP on end errors and peak speed.

There are multiple mechanisms of plasticity at the PF-PC synapses [46]. We explored the hypothesis that a dual STDP process in PCs, comprising error-dependent LTD (Eqs 5.3–5.7) and error-independent LTP (Eqs 5.2–5.3), is involved in decreasing the end foveal error and increasing the peak speed of eye movements. Briefly, LTP potentiates the PF-PC synapses, and therefore it increases PC burst activity above the baseline firing rate, while decreasing PC pause. Conversely, LTD happens in response to the presence of the error signal, conveyed by the climbing fibers. Thus, it decreases the PC burst proportional to the end foveal error, while it increases the drop of PC pause below baseline. In combination, the LTP and LTD can set appropriate firing rates in both burst and pause PC subpopulations, to control the overall motor drive output from the cerebellum (sent as a speed command to the brainstem burst neurons).

To test our hypothesis, we embedded a spiking cerebellum model, derived from [42–44] in a saccade controller, derived from past literature on brainstem neural response characteristics [16] (for details see *Relationship between the model cerebellar activity and movement control signals* in the Methods section). In our model, the cerebellar output transmitted by the Deep Cerebellar Nuclei (DCN) modulates the input firing rate to the brainstem burst generator, which subsequently sends speed commands ($u$) to the downstream premotor neurons and oculomotor circuit (Fig 1A). The speed commands were further transformed into oculomotor actuation by the neural integrator pathway (simulated as a parallel proportional integrator, see Fig 1A and [16]). The range of foveal error magnitudes experienced in a given simulated eye movement determines if the PF-PC synapses undergo LTP or LTD, as illustrated in Fig 1B. The inferior olive (IO) firing rate, depends on the end foveal error and affects the amount of LTD occurring in the PF-PC synapses, while there is a constant LTP process that is active independently from the error. As shown in Fig 1B, if the errors are too high (>1 degree), then the IO displays a high probability of spiking, which, after multiple trials, causes a higher LTD compared to LTP. When the error is smaller (between 0 and 1 degree), the IO spiking probability is proportional to the error magnitude that can have different STDP effects on burst and pause PC subpopulations. Such a linear, saturated IO firing rate as a function of end foveal errors is also observed in the probability of CSpikes occurrence (caused by changes in the IO spiking probability) when monkeys make erroneous movements [33].

We simulated saccades towards a fixed spatial target that required 10-degree eye rotation. In the absence of cerebellar contribution, the base controller (composed of a proportional burst generator) driven by the target displacement signal and an internal feedback loop (see Fig 1A) made the eye overshoot the target by ~3 degrees, while generating low speed movements with a peak speed of 383 deg/sec and a duration of 62 ms (Fig 2D and 2E). We then expected that the cerebellum could leverage its dual STDP processes at the PC synapses to gradually decrease the end foveal error and increase the peak speed over repeated eye movements to the given target.

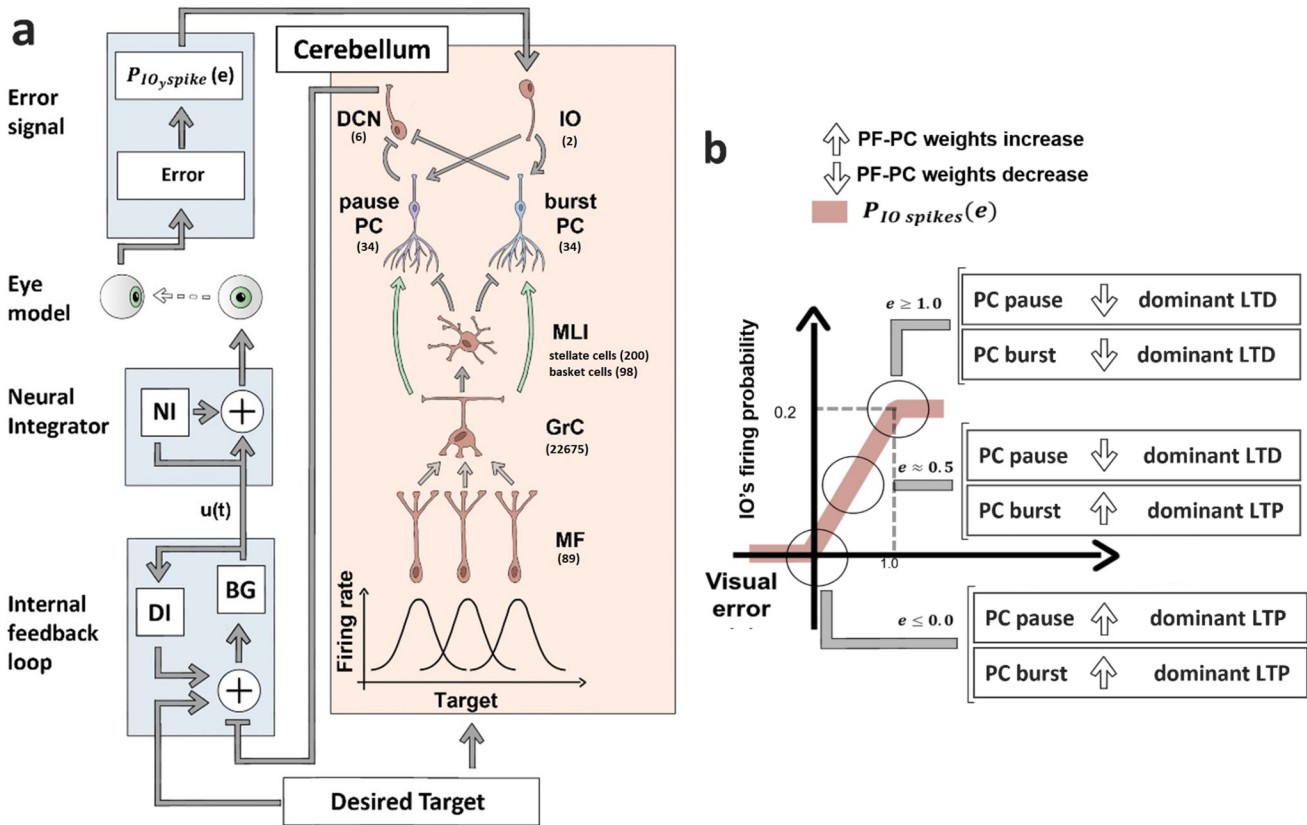

**Fig 1. Schematic of the control loop.** a) The target displacement information (desired target) is sent to the internal feedback loop (IFL) and to the cerebellum. In the cerebellum, the target displacement is encoded by the gaussian receptive field of the Mossy fibers (MF), MF are connected to the granule cells (GrC), which, through their axonal endings (namely the parallel fibers (PFs)) excite both the Purkinje cells (PC), and the molecular layer interneurons (MLI), composed of basket and stellate cells. The connection between PF and PC is the only plastic one in the model (represented by the green arrows). The MLI are connected to the PCs. The PCs are split in two subpopulations: pause PC and burst PC (light blue). PCs are connected to the deep cerebellar nuclei (DCN), which are the output of the cerebellum and project to the IFL. The IFL is composed by the Displacement integrator (DI) and the Burst generator (BG), and sends signals to the neural integrator (NI). The IFL generates speed command (u(t)) to be followed by the eye, while the NI transforms this speed command into motor torques by means of pulse-step integration. If the resulting movement is erroneous, then the error information is encoded by the firing rate of the inferior olive (IO), which projects onto to the PCs. b) Relation between the foveal error (e) and the IO firing probability (PIO spikes). The effect of different PIO spike on the PF-PC weight changes are shown for 3 scenarios for the foveal error: error < = 0, error = 0.5 and error > = 1. For each error range, the effect of the related IO's firing probability (producing an LTD in PF-PC synapses), and the error-independent LTP on the two PC subpopulations is showed. If the arrow points down the overall PF-PC weight decreases due to dominant LTD process, decreasing the firing rate of the PCs on subsequent trials, if the arrow points up then the resulting weight change is positive due to dominant LTP process, increasing the PC firing rate on subsequent trials.

Indeed, a cerebellar spiking neural network learning by means of dual plasticity rule led to decrease in end foveal error and increase in peak speed across repeated movements. Comparing movement kinematics before and after cerebellar training (Fig 2D and 2E) revealed that the movements were much faster after concluding cerebellar training (536 deg/sec) compared to the initial learning period (448 deg/sec). Note that the untrained cerebellum (before learning) network was initialized with low PF-PC excitatory weights to minimize the cerebellar influence on the brain stem motor drive. The movement duration decreased from 62ms without cerebellum to 30 ms after cerebellar training. Notably, Fig 2G shows that the error is reduced quickly from 2.6deg to 0.5deg within 25 trials, while the peak speed increased slowly (Fig 2F).

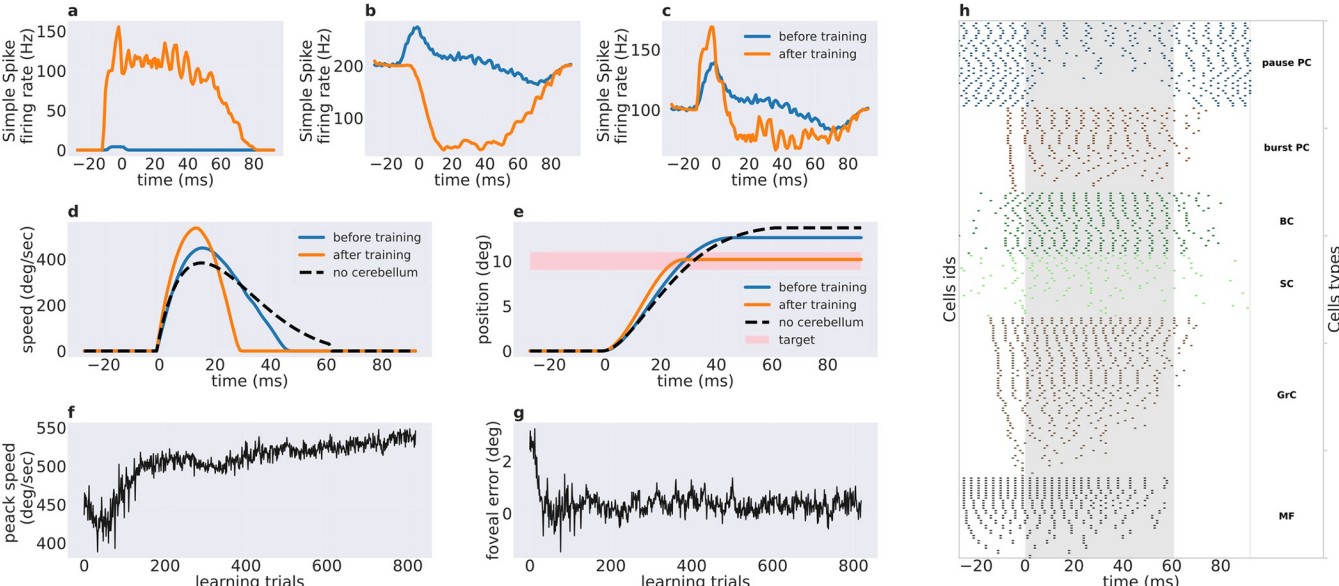

**Fig 2. Effect of cerebellar learning by dual PF-PC plasticity mechanisms on the behavioral and neural variables.** Simple spike firing rate: **a)** burst PC subpopulation. **b)** pause PC subpopulation, **c)** all PCs together. In these panels, the simple spike activity before training is shown in blue while the one after training is shown in orange. t = 0 ms corresponds to the onset of the eye movement. Eye movement kinematics: **d)** saccade speed and **e)** displacement of the eye model before training, after training and without cerebellum in blue, orange, and dashed-black. The pink rectangle represents the target within accuracy limits of +-0.5 degree. t = 0 ms corresponds to the onset of the eye movement. Movement properties across trials: **f)** Peak speed, **g)** end foveal error. **h)** Raster plot of the spiking activity of all cerebellar neuron populations in the model. The dots correspond to a single spike of the given cell, where the cell type is denoted in the right column. The eye movement starts at t = 0 ms and the gray box highlights the extent of movement after training. While all the pause PCs and burst PCs are shown, for the other populations only the cells that spike the most during the trial are shown.

We examined the spiking activity of PC populations (SSpike) before and after cerebellar learning by the dual-plasticity rule (Fig 2A–2C). As described in Methods, the PCs are comprised of burst and pause PC subpopulations. We computed the total firing rate across PC neurons within both "burst PC" and "pause PC" subpopulations. The burst and pause PC subpopulations showed significant and mutually opposite changes in firing rate after training. The burst PC subpopulation activity showed an increased SSpike activity by a maximum of 155 Hz compared to the baseline (see Fig 2A). The pause PC subpopulation reduced its activity by 164 Hz from its baseline level (see Fig 2B). The simulated pause PCs show a baseline SSpike firing rate of 200Hz, whereas the biological recordings show 50-100Hz SSpike rate (see supplementary material in [33]). This discrepancy arises from the need to reduce computing time by constraining the maximum number of PC neurons. Notably, similar results can be obtained when the baseline firing rate is tuned at realistic SSpike levels of 50-100Hz, but require larger number of neurons to produce the same motor output range. For both burst and pause PC subpopulations, the firing rate was significantly different compared to their corresponding pre-training levels (p <0.001 paired t-test, considering the first and last 10 trials, where plasticity is switched off; normality was checked for the first and last 10 trials, pval of the kstest, respectively, 0.58 and 0.90). After training, the burst PC subpopulation anticipated movement onset by 15ms, while the untrained burst PC subpopulation showed very low burst activity with a small bump close to 15ms similar to the trained subpopulation. Notably, after training, the burst PC subpopulation activity increased during both the anticipatory period and during movement, showing that the LTP modulates the population activity in both task periods.

In aggregate, after training, both the burst and pause PC subpopulations (Fig 2C) showed an initial increase in firing rate, reaching the overall peak activity by before the movement

onset. Then, the firing rate dropped below the baseline that was recovered again after 80 ms, (i.e. beyond the time at which the movement ends at t = 30 ms). Notably, the overall PC population activity peaked 25ms earlier (around t = -10 ms) than the eye speed (around t = 15 ms). The shape of the PC population signal, as well as its phase relationship with the eye speed, bear similarity with experimental observations [32]. While monitoring the trial-by-trial activity modulation, the burst PC subpopulation started with low activity and gradually increased its response in both the anticipatory period (t<0) and the movement period (t > 0) with training, while the pause PC subpopulation started with a slight burst in the anticipatory period in the early training trials (S1 Fig). Along trials, this anticipatory burst was reduced and the firing rate dropped below baseline during the movement and post-movement periods (t > 0, until t = 87 ms).

The spiking cerebellar model was built based on previous cerebellar microcircuit models [44] to emulate the spiking properties of multiple cell types with realistic connectivity. Hence, each neuronal population expresses specific spiking patterns. After concluding cerebellar training, all cerebellar populations showed activity spanning both preparatory, movement and after-movement periods (Fig 2H). In summary, the dual plasticity at PC synapses can increase-peak speed and decrease the end foveal error, by acting on subpopulations with specific time windows and direction of modulation relative to baseline firing rate.

## Saccades to different target amplitudes display main-sequence relationship

Saccades in humans and monkeys are characterized by stereotypical relationship between peak speed, duration and amplitude of movement [47, 48] known as the "main sequence". Here we evaluated if synaptic changes occurring at the level of cerebellum can emulate such stereotypical patterns, while the target related information from the MFs remain unaltered across trials. We trained the cerebellum spiking neural network to make saccades to multiple target locations on horizontal axis, while keeping the desired target signal and the brain stem circuit parameters fixed throughout the training.

In Fig 3 we show the relationship between amplitude and peak speed, duration of the emergent saccades after training with two different variations of the cerebellum model. The relationship between peak speed, duration and amplitude that included dual-plasticity in the cerebellar model is comparable to that of monkey eye movements (see Fig 5 in [48]). On the other hand, the cerebellum model with only-LTD produced relatively slower saccades with longer duration. We plotted the trial by trial changes in end error and peak speed across training trials sorted according to the four different target locations in S2 Fig. Note that both dual-plasticity and only-LTD variations were initialized with the same synaptic strengths at each synapse before training, and hence generating same kinematics at the very first trial. Both dual-plasticity and only-LTD models generated lower errors with training, but differed in their overall kinematic solutions during and after training. Only-LTD model decreased the error by reducing the peak speed across trials compared to the initial peak speed. On the other hand, dual plasticity model showed the capacity to increase peak speed across trials once a sufficiently low level of error is reached. The difference between both conditions was due to presence of LTP in the PF-PC synapses, which increases the total PC population firing rate above baseline across trials (as shown in Fig 1), hence providing a higher motor drive in the early movement period to increase the peak speed.

## PC population activity predictively encodes eye speed

An important experimental observation is that the PC population activity predictively encodes eye speed: when making saccades to a spatial target at a fixed displacement, the activity peak

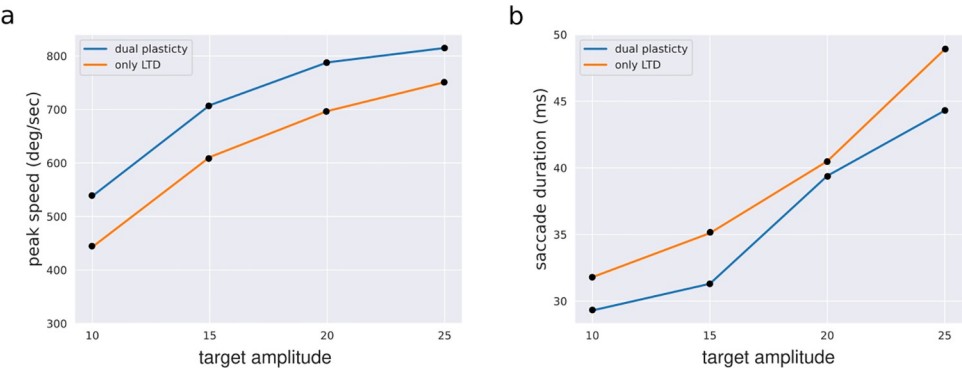

**Fig 3. Exemplary saccade kinematics across different target amplitudes.** a) Peak speed against target amplitude. b) saccade duration against target amplitude, tested at four different target amplitudes at 10, 15, 20, 25 degrees using cerebellum models with dual plasticity and only LTD.

was linearly related to the speed peak and preceded the actual movement [32]. Does the PC population activity emerging from our model match these recent experimental observations from primates?

We investigated how changes in PC population activity induce changes in eye movement. To do this, we forced the modulation of PC activity by changing the cerebellar network input at the Mossy Fibers (MF), and then we compared the amount of change in the PC population activity with the change in eye speed. Note that here we modulated the MF activity to produce variation in the PC output, instead of directly modulating the PC activity by adjusting the voltage dynamics of the PCs. This protocol allows us to understand if the PC output and the eye speed can be controlled by task relevant inputs through MF afferents (projecting from regions upstream of the cerebellum) driving the cerebellar activity.

We systematically decreased the total input to the trained cerebellar spiking network and examined the impact on the PC activity and eye movement. The cerebellar model receives inputs from MFs, which excite the GrCs which ultimately influencing the PCs through direct (GrC-PF-PC) and indirect (GrC-PF-MLI-PC) pathways. Both burst and pause PC subpopulations showed modulation in SSpike activity when the MF input-level was reduced (100%, 83%, 67%, and 50% of the standard input used for training). The burst PC subpopulation showed a decreased SSPike activity, characterized by a reduction in peak firing rate from 148 Hz to 103 Hz with half MF input. Moreover, the burst duration decreased along with reduced input-levels due to both a later onset and an earlier burst ending (Fig 4A). On the other hand, the pause PC subpopulation showed a decrease in the pause depth from 160 Hz to 85 Hz with half MF input (Fig 4B). As a whole, the total PC population activity showed a lower peak firing rate during the anticipatory period (t < 0ms) and a less pronounced pause in the movement period, when the MF input-level was progressively reduced (Fig 4C).

Finally, we compared the total PC population activity with the corresponding eye speed in the different input scenarios (Fig 4D). The PC activity predictively encoded and controlled the eye movement. Indeed, the peak activity always preceded the speed peak, and it was linearly correlated to the speed peak (Pearson correlation = 0.99, p value < 0.01, Fig 4E). Potentially, the reduction in PC activity might not only reduce the speed, but also the increase the end foveal error. In this case, the movement might have occurred at lower speeds simply because the eye displacement incurred a higher foveal error, instead of a direct relationship of eye speed with the PC activity. To check this aspect, the end foveal errors were plotted in the different input scenarios. The foveal error was similar across the simulated input levels, thus across

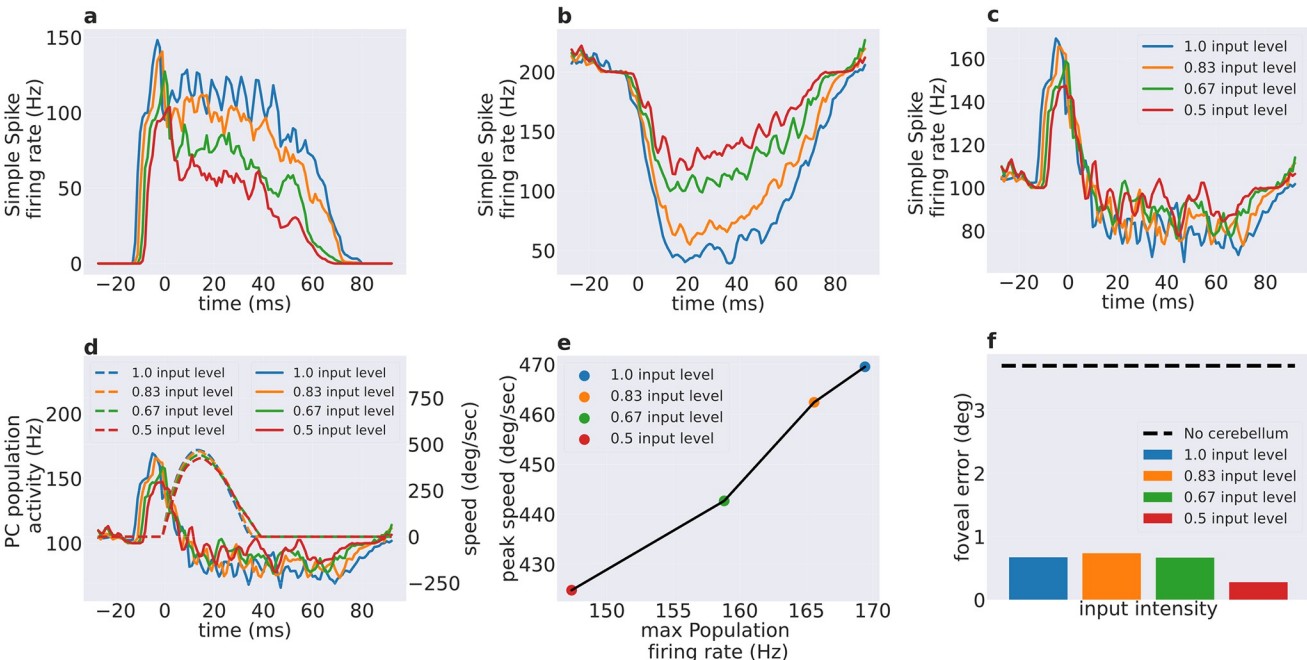

**Fig 4. Predictive encoding of saccade eye speed in PC activity.** Modulation of simple spike firing rate caused by decreased input on the cerebellar MFs (1, 0.83, 0.67 and 0.5 of the input used for training) for **a)** the burst PC subpopulation, **b)** the pause PC subpopulation, and **c)** for all PCs together (t = 0 ms is the onset of the eye movement). **d)** Comparison of PC population activity and saccade eye speed (dashed lines) for different MF input levels. **e)** Relationship between the peak of PC SSpike firing rate in the anticipatory period and the corresponding peak eye speed. Pearson correlation = 0.99, p value < 0.01. **f)** The foveal error occurring at the different input levels to the cerebellar network. The dashed black line corresponds to the foveal error without the cerebellum.

the PC activity levels. The movement error was ~0.5 deg in all scenarios without any significant differences (Fig 4F). Therefore, the reduction in PC activity resulted in decreased eye speed, while the end foveal error remained unchanged. To compare the effect of input level on the maximum PC population activity and maximum speed, two separate one-way ANOVA analysis were performed. Each input level had 10 separate saccades towards the 10-degree target (while the PF-PC plasticity was inactivated to have identical condition of movement). The one-way ANOVA revealed that there is a significant statistical difference in the maximum PC population activity across input levels ($F_{(3,36)}$ = 103.15, p < 0.001), and similarly significant statistical difference in the peak speed across input levels ($F_{(3,36)}$ = 133.26, p < 0.001). Post hoc comparisons with paired t-test were used to contrast the maximum of the population activities and the maximum speed obtained with the different input levels. For population activity, significant differences were found between each pair of input levels (p<0.001 after Bonferroni correction) except between input level = 1.0 and input level = 0.83 (p = 0.18 after Bonferroni correction). For the maximum speed, significant differences were found between each pair of input levels (p<0.01 between input-level = 1.0 and input-level = 0.83, and p <0.001 for remaining pairs after Bonferroni correction).

## Each plasticity component at PC synapses distinctly controls one aspect of the saccadic movement

How each component of PF-PC plasticity ('error-independent LTP' term and 'error-dependent LTD' term) influence the neural circuit dynamics and the consequent movement parameters during saccade is still unclear. Here, the cerebellum network was trained during repeated

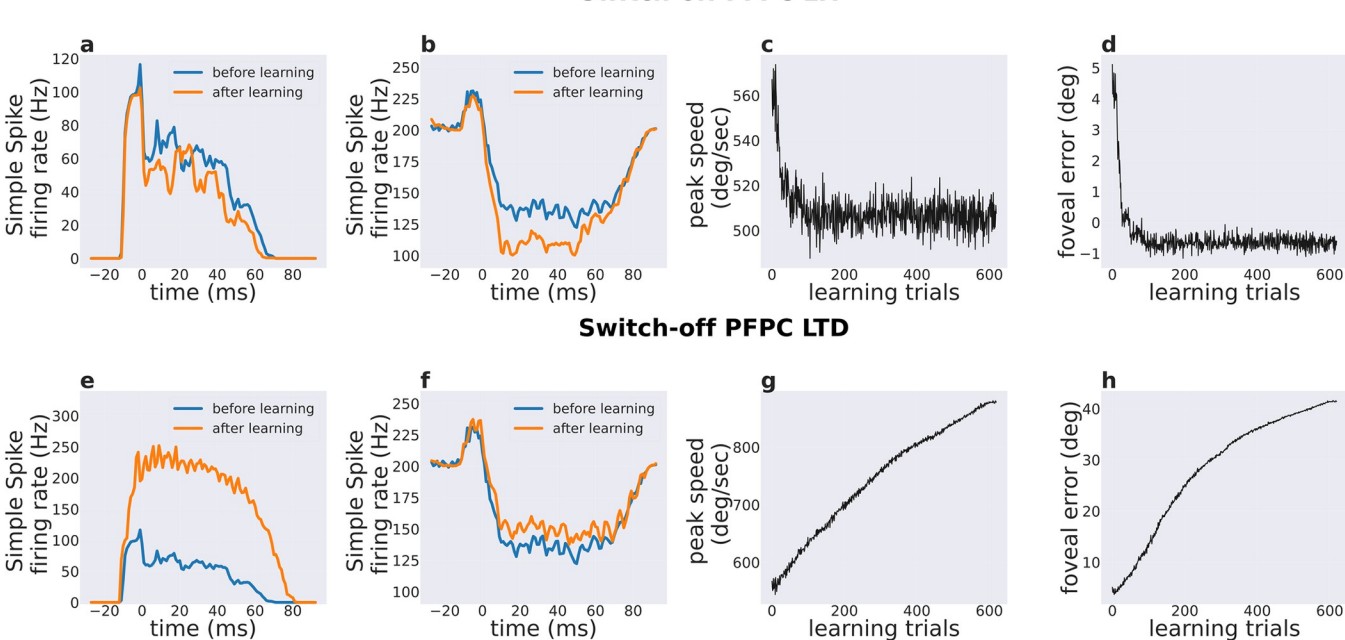

**Fig 5. Effect of selective switch-off of plasticity (LTP or LTD) on PC population activity and saccade kinematics.** Simple spike activity before and after training when switching-off LTP or LTD. in a,e) the burst PC subpopulation and in b, f) the pause PC subpopulation. c, g) Peak eye speed across learning trials when switching-off LTP or LTD. d, h) Foveal error across training trials by switching-off LTP or LTD.

saccades to reach a target requiring a 10-degree eye rotation, by switching-off one plasticity (LTP or LTD) in turn during training. To better assess the effect of blocking LTP without losing generality, we used a different set of initial pre-training PF-PC weights compared to the original one (cf. Fig 2).

Switching-off LTP resulted in a rapid decrease in eyes peak speed to 515 deg/sec around the 27th trail, followed by a constant peak speed at 505 deg/sec. These eye speeds were significantly lower compared to the simulated healthy saccades with a peak speed of 536 deg/sec (Fig 5C). The movement end error quickly reduced from 5.1 degree to 0 degree within 27 trials even though LTP was absent (Fig 5D). Interestingly, the switching-off of PF-PC LTP did not allow to maintain a constant peak speed across training trials, and instead caused a reduction in peak speed with repeated movements. This indicates that LTD (in the absence of LTP) reduced peak speed to decrease the foveal error, which is an intuitive solution if no constraints are placed on movement speed.

The burst PC subpopulation firing rate changed in a different manner when LTP was blocked compared to the normal training scenario that has been presented earlier in Fig 2. Specifically, the burst PCs decreased their activity with training in both anticipatory and movement periods (Fig 5A) (p <0.001 with paired t-test, considering the maximum burst activity in 10 trials each before and after training; normality was verified for both groups of the data using kstest). On the other hand, pause PC subpopulation increased the amount of firing rate modulation after training (Fig 5B) throughout the movement period (p <0.001 with paired t-test, considering the minimum activity in 10 trials each before and after training); normality was verified for both groups of the data using kstest. Note that the changes in both burst and pause subpopulations activity was determined by the LTD only, indicating that LTD influences responses in both PC subpopulations.

Switching-off LTD plasticity resulted in a gradual increase of the speed peak, driven by LTP, and lead to high errors (Fig 5G and 5H). This essentially means that LTD is necessary to counter-act the unstable behavior caused by LTP, as LTP did not depend on movement errors for updating the PF-PC synaptic weights. Switching off LTD caused opposite effects on the activity of burst and pause PC subpopulations, compared to those observed when LTP was switched-off. The activity of burst PCs increased above the baseline (Fig 5E) (p <0.001 with paired t-test, considering the max activity in 10 trials each before and after training; normality was verified for both groups of the data using kstest). On the other hand, the activity of pause PCs decreased towards the baseline (Fig 5F) (p <0.001 with paired t-test, considering the minimum activity in 10 trials each before and after training; normality was verified for both groups of the data using kstest).

Overall, the results from selective switch-off of LTP or LTD show that LTD is responsible for increasing movement accuracy by decreasing the end foveal error, while LTP is responsible for increasing movement peak speed.

## Blocking plasticity at either burst or pause PC subpopulations impairs the increase in eye speed with training

Is the dual-plasticity mechanism at both burst and pause PC subpopulations necessary for improving accuracy and peak speed of saccades? To this aim, the cerebellar network was trained during saccades to a 10-degree target, by switching-off the plasticity at either burst or pause subpopulations.

When dual-synaptic plasticity was switched-off only at the burst PC subpopulation, both end foveal error and peak speed decreased rapidly till the 49th trial, after which error reached low values close to 0 degrees and the peak speed remained constant around 500 deg/sec (Fig 6C and 6D). Notably, the peak speed was much lower after training compared to the initial

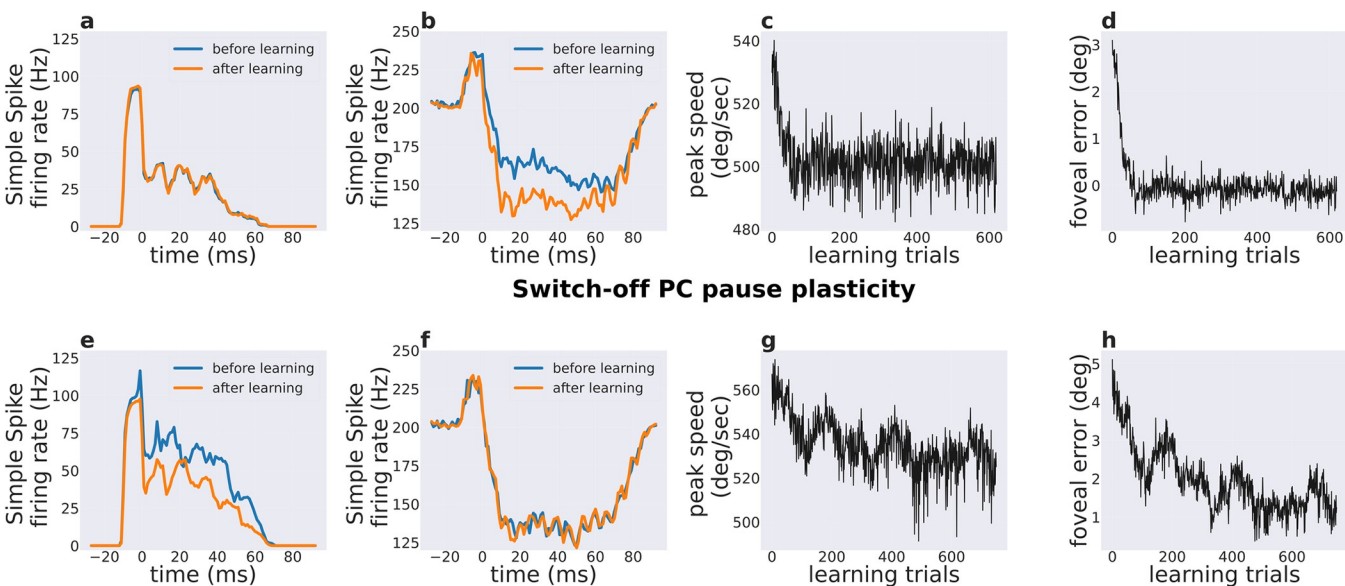

**Fig 6. Effect of selectively switching-off of dual plasticity process in each PC subpopulation on the PC population activity and saccade kinematics.** Simple spike activity before and after training in a, e) the burst PC subpopulation when switching-off burst or pause PC plasticity and in b, f) the pause PC subpopulation when switching-off burst or pause PC plasticity. c, g) Peak eye speed across learning trials when switching-off burst or pause PC plasticity d, h) Foveal error across training trials by switching-off burst or pause PC plasticity.

value of 530 deg/sec. As expected, the burst PC subpopulation firing rate remained constant (Fig 6A) without statistical significance (p = 0.16 with paired t-test, considering the maximum activity in 10 trials each before and after training; normality was verified for both groups of the data using kstest). Pause PC subpopulation firing rate showed statistically significant modulation during the training (Fig 6B) (p <0.001 with paired t-test, considering the minimum activity in 10 trials each before and after training; normality was verified for both groups of the data using kstest).

Even when the dual-synaptic plasticity was switched-off only at the pause PC subpopulation, both end foveal error and peak speed decreased during training trials (Fig 6G and 6H). Therefore, switching-off the dual-synaptic plasticity on one of the two PC subpopulations impacted similarly on the overall saccade kinematic; however, the rate at which the error and peak speed decreased was more than when blocking the burst PC plasticity (Fig 6G and 6H). As expected, when the dual-plasticity is switched-off at the PC pause subpopulations the population firing rate remained constant (Fig 6F) (p = 0.23 with paired t-test, considering the minimum activity in 10 trials each before and after training; normality was verified for both groups of the data using kstest). However, the burst PC subpopulation firing rate decreased during training (Fig 6E) (p <0.01 with paired t-test, considering the maximum activity in 10 trials each before and after training; normality was verified for both groups of the data using kstest). This is different from the case when both burst and pause subpopulations had active plasticity, where the burst subpopulation showed an increase in firing rate with training, whereas the pause subpopulation decreased firing rate below baseline during movement and post-movement period. Hence, the dual-plasticity rule in the model flexibly influenced the PC firing rate modulation depending upon whether one or both burst and pause PC subpopulations underwent plasticity. Notably, such flexibility was not hardcoded for our saccade simulations but rather it emerged from the interaction between LTP and LTD in the burst and pause PC subpopulations.

In summary, the movement accuracy gradually improved even if the dual plasticity was limited to only one of the burst and pause PC subpopulations. On the other hand, the absence of plasticity in either one of the PC subpopulations led to slower eye movements. When plasticity was switched-off at the burst PC subpopulation, the total change in PC population activity was dominated by a decrease in pause firing rate below baseline, subsequently leading to slow eye movements. When plasticity was switched-off in the pause PC subpopulation, the firing rate of the burst PC subpopulation after training decreased, while the burst firing rate increased when synapses converging onto both subpopulations were plastic. The decrease in burst firing rate resulted in slow (but still accurate) movements, while the pause PC firing rate remained constant around the pre-training level.

## Discussion

The main result of this study is that millisecond-precise spike timing in PCs is needed to implement an STDP learning rule allowing the anticipatory control of saccadic eye movements. Computationally, this is not trivial, since STDP controlled both error and peak speed of saccades, while the total MF pattern of activity remained unchanged across trials, even though PCs receive only end foveal errors as the evaluative information. In the model, STDP is comprised of LTP and LTD at the PF-PC synapses providing suitable proxies for the hypothesized plasticity mechanisms: while LTD, driven by the foveal error, adjusts the cerebellar output such that errors are rapidly decreased in subsequent movements, LTP occurs independently from the foveal error to slowly increase the movement peak speed. Eventually, the emergent population activity shows predictive encoding of saccadic eye speed by modifying the burst/pause patterns of PCs.

The existence of multiple plasticity mechanisms at PF-PC synapses is well known from experiments. A most renowned form of plasticity is based on the co-occurrence, in PCs, of signals coming from parallel fibers and climbing fibers originating from the IO. Briefly, when an unexpected situation occurs such as pain, retinal slip or movement errors, the activity of IO neurons generates CSpikes in PCs. The increase in PC CSpikes subsequently enables the induction of LTD in the parallel fiber synapses, whose activity coincides with the CSpikes [21, 39]. There is also another kind of plasticity in PF-PC synapses called long-term potentiation (LTP) [37, 38, 49], whose putative function is to prevent the synapses from saturating due to LTD and, as a consequence, to maintain the plasticity capability of the cerebellar circuit [50]. LTP is shown to occur in the PF synapses that are active in disjunction with CSpike activity [51]. In our model, we showed that LTP serves a distinct behavioral role that optimizes motor behavior, by increasing the movement peak speed, in addition to preventing the neural responses from saturating. While the previous experimental studies indicate why it is important to have bi-directional STDP to maintain active plastic synapses [52], our results show the relevance of having multiple plasticity mechanisms to control different parameters of motor behavior such as movement error, peak speed and duration.

In this work, we explored the function of two among multiple plasticity mechanisms in the cerebellar microcircuit [46, 53]. Even the PFs project onto the PCs through two different pathways, direct excitatory PF-PC connections and di-synaptic inhibitory PF-MLI-PC connections [54, 55]. In the presented simulations, we considered the PF-MLI-PC synapses to be fixed and simulated the effect of STDP in direct PF-PC connections only. We expect that similar effects can be reproduced by means of PF-MLI-PC plasticity.

In [25, 26] the authors showed that during saccade adaptation to abrupt target shifting, the new eye trajectories exhibited different patterns of change in early and late movement periods (corresponding to the acceleration and deceleration phase, respectively). If different timescale processes were used to adjust the motor commands for acceleration and deceleration periods within a movement, then one might expect dissociation between changes in peak speed and end displacement (and hence end movement error). Recently in [25], the authors showed that the motor commands in the acceleration period change relatively slowly compared to those in the deceleration period across saccade adaptation trials. A slower change in the accelerating motor command can produce slower increase in peak speed, while a faster change in decelerating command can ensure that the saccade movement errors are quickly minimized. Our model predicts that the changes in peak speed can be mediated by error-independent LTP, which produces synaptic changes at a slower time-scale compared to error-dependent LTD. Open questions for network modeling are now whether and how different plasticity mechanisms within the cerebellar circuit could explain the systematic decay of adapted motor commands during pauses/breaks between different blocks in the experiment and error clamp conditions as observed in [25].

While we have shown that the dual STDP process improved saccadic learning and control, the mechanisms that ensure a proper coordination between the underlying LTP and LTD components remain to be characterized. In-vivo experimental recordings [33, 34] of PC activity suggest that changes in CSpike activity is followed by changes in the PC SSpike frequency in successive movements. [34] observed that the CSpikes show a characteristic pause (inhibition of CSpikes) during movement in addition to the well-known modulation after the end of eye movement. The CSpikes were paused for the entire duration of movement, in a way that was independent of the end result of eye movement and hence potentially independent of any resulting visual error. By considering in vitro experimental and computational observations of [30, 51], a reduction in CSpike probability should result in LTP at the PF-PC synapses. Thus, if

the CSpikes pause for the duration of the movement, then LTP should depend on movement duration i.e., the larger the movement duration the larger LTP. In this study, we show that a larger error-independent LTP increases the eye speed. However, if the eye speed is uncompensated, then the foveal error increases along with the CSpike frequency in the post-movement period. This enhances LTD and, eventually, reduces the error itself. In aggregate, if CSpikes would depend on multiple types of information (i.e. not uniquely on movement error), the dual STDP mechanism could optimize the saccade as a whole rather than just reducing the foveal error. Note that by using term 'optimize', we indicate movements that show high peak speed and low movement duration at a given target displacement, while reaching the target accurately i.e., with low foveal error. Indeed, a recent experimental study [56] showed that CSpikes carry information about multiple parameters of primary and corrective saccades in a time-specific manner. Specific investigations on the information content of CSpikes (for example [57]) may unravel which aspects of motor behavior can be optimized by the cerebellum, and how STDP accounts for an abstract behavioral goal as a whole rather than simply compensating a pure visual error.

By taking the case of saccadic adaptation, this study *de facto* addresses the fundamental question of how the cerebellum optimizes motor behavior. The optimality of a movement depends upon various factors such as the incurred kinematic errors, energy consumption, peak speed, duration etc., and is generally expressed as a cost function that quantifies how closely do the actual movement variables follow a reference trajectory or a movement goal [2, 26, 58]. Earlier works on cerebellum mainly focused on reflex behaviors such as vestibulo ocular and optokinetic reflexes (VOR/OKR), where the reference trajectory is available in terms of physiological signals encoding retinal slip [29, 41, 59–61]. If the optimization is formulated in terms of minimizing retinal slip as the cost and cerebellar PF-PC weights as the adjustable parameters, it was earlier shown that cerebellum-like synaptic plasticity rules emerge [31, 55]. Notably, continuous information about retinal slip was required for such plasticity to properly optimize the movements. In saccades, such a reference signal does not exist, as it is not possible to know the optimal movement duration or speed before the actual movement begins [58, 62], hence requiring "*unreferenced optimization*". It has been hypothesized that the cerebellum might be involved in optimizing a cost function based on movement duration, where the cerebellum controls when a movement should end (see [12, 63]), but the underlying circuit principles behind such optimization of movement duration remain unknown. In a simplified mathematical model, earlier we showed in [35] that unreferenced optimization of saccades in the cerebellum should rely on dual-plasticity processes: one that depends on the foveal error, and the other that is a function of the total PC firing rate during the eye movement. Here, a more realistic setup exploits PF-PC plasticity expressed in terms of STDP learning rules. In our simulations (Fig 2), we observed that the foveal error rapidly decreases within a few trials while the peak speed increases at a relatively slower rate. This is an important feature for model validation, since a recent behavioral study discovered a similar dual regime for trial by trial changes in speed and visual error during saccade adaptation [64]. Our model also predicts another relevant feature: once a low foveal error is quickly attained, the eye speed slowly keeps increasing until small movement errors are generated and engaging the error-dependent plasticity (the errors can be very small). Subsequently, the error-dependent plasticity dominates the peak speed term changing the PF-PC weights and restoring the accuracy. Overall, our model shows how simple STDP rules at the PF-PC site can flexibly combine to perform unreferenced optimization.

Notably, the PF-PC plasticity mechanisms in this model are the same as used in previous models to simulate a variety of tasks including VOR adaptation, force-field adaptation, eyeblink conditioning adaptation, object manipulation [41, 42, 65]. Therefore, we believe that the

simulated plasticity mechanisms are well suited to control a large set of movements when combined appropriately and applied to sensorimotor signals encoded in different ways, supporting the tenet that the cerebellar circuit implements a general computational algorithm that can be applied to a number of different cases depending on the specific wiring with other brain subsystems [30, 44, 66, 67].

The present saccade control circuit is built on top of previous successful implementations of saccade behavior, with an emphasis on the STDP processes of the cerebellar circuit while approximating the extra-cerebellar regions with a lumped control theory-based implementation. The extra-cerebellar loop and the oculomotor dynamics is similar to that used in [16, 18]. Beyond initial models proposed for simulating the role of cerebellum in saccade adaptation [19, 20, 68], a more recent study [22] for the first time proposed the relevance of using separate burst and pause PC subpopulations. The predictions from [22] have been addressed experimentally by [32], showing the symbiotic relationship between detailed network simulations and experiments. A plausible biological counterpart of the multiple PC types (based on burst and pause SSPike behavior) and multiple STDP mechanisms used in our simulations, is related to the Z+/Z- stripe-based segregation of PCs and to bidirectional learning in upbound and downbound microzones of the cerebellum [69]. We expect that future experiments will test our model predictions to show that error-independent LTP is required for improving peak speed, and how dual synaptic processes interact with the burst-pause PC subpopulation in controlling saccade adaptation.

In conclusion, the saccade is a prototype of a ballistic movement fully operating in forward mode (there is no time for online sensory feedback until the saccade is terminated, hence the need for pre-programming using the end-point error for learning). Movement is programmed by the brain as a sequence of single jerks [70, 71], each one representing a tiny ballistic movement. Therefore, this work highlights the basic mechanisms through which the cerebellum could control complex movements generated as sequences of apparently continuous movements. For example, playing piano can be assimilated to a sequence of ballistic fingers movements, each one preprogrammed and concatenated with the previous and the next ones. It will be interesting to see whether and how the same cerebellar operations observed in saccades operate on movement sequences.

The limitation of the current model is the simplicity of different brain regions, in spite of a detailed biologically inspired construction of the cerebellar network. The model focused on how the PC population activity can be adaptively modified to influence peak speed, and end visual error while the target-related input drive to the cerebellum and the rest of the controller remains unaltered across trials. But the saccade control is composed of different regions, and especially Superior colliculus (SC) activity is implicated in determining a motor plan, that the downstream cerebellum and brainstem burst neurons can implement [72, 73]. In principle, the SC activity can be altered to change the input drive (MF activity) to the cerebellum and the brain stem burst neurons, which can subsequently lead to change in peak eye speed. Notably, our modeling results do not negate the possibility that the SC influences peak eye speed by modulating its input MF drive to the cerebellum. Indeed, in Fig 4, we show that modulating the input MF drive to the cerebellum results in different peak eye speeds. However, the set of presented results are better viewed as evidence that the cerebellar STDP mechanisms can exert peak speed modulation, even if the SC/target-related input drive remains constant (see [27]). Furthermore, our future work will focus on implementing a more realistic DCN circuit. Each DCN is a hub of diverse incoming projections from MFs, PCs, and also a recurrent nucleo-olivary inhibition loop [74]. Biological DCNs have additional connections from MFs and the nucleo-olivary loop that are not included here, potentially limiting the predictive capabilities of our model. Functionally, the DCNs can display unique spiking characteristics such as

rebound spikes during saccades [22], that can help them sustain motor output while the inputs are extinguished. The DCN-IO circuit is important to generate internal spatio-temporal dynamics coordinating large sets of PCs [46] and to regulate plasticity in the circuit by adding slower time constants to the learning process [40, 53, 75]. These aspects should become critical in complex multifactorial behaviors evolving over multiple spatial scales but, since this was not the case here, the absence of a more precise representation of the IO-DCN loop was unlikely to cause relevant drawbacks. Overall, it remains to be seen how the cerebellum model presented in this work shares computations with other regions, by integrating realistic circuit models of SC, DCN, and brainstem circuits.

## Materials and methods

### Control loop

We used the same control loop described in [16, 17]. The control loop is split into three components (see Fig 1A). The first one is the physical eye model, as described in [16], the second component is a brainstem-like internal feedback loop (IFL), the third component is the cerebellum which compensates for erroneous movements generated by the brainstem.

   In our saccade control loop (depicted in Fig 1), the desired target displacement of the eye is represented using rate-based encoding, where 1 degree = 1 Hz. In other models, a more complex temporal signal particularly describing superior colliculus activity has been used as target information [18]. However, since our focus is on the cerebellar contributions to saccade adaptation, we chose the simple rate-based encoding as it can reproduce movement kinematics at different targets while avoiding the complexity associated with generating moving-hill waveforms, specific to the superior colliculus [72, 76] (not included in this model). Overall, as shown in Fig 1, there are two parallel pathways to transform the selected visual target to motor command generation. The direct pathway (left part of Fig 1A) that performs target-motor command mapping through the IFL (putatively in the brainstem), and the indirect pathway (right part of Fig 1A) that involves cerebellar modulation of the IFL activity.

   Overall the IFL outputs a speed command to the eye, which is subsequently transformed into motor activation through downstream neural integrator circuit. If the eye does not reach the target with the given motor activation, the difference between the target location and the eye displacement at the end is detected as end foveal error. This foveal error then influences the IO firing rate (as shown in Fig 1B), subsequently leading to updates in the weights of the PF-PC connections inside the cerebellum, to decrease the foveal error in subsequent trials.

### Eye model

The eye model is described as a linear dynamical system:

$$\dot{x(t)} = A\, x(t) + B\, u(t) \tag{2.0}$$

   Where $x(t)$ is a 2-elements array of position ($p(t)$) and speed ($v(t)$) at time $t$ where $x(t) = [p(t), v(t)]$. $u(t)$ is the brain stem burst generator output, which provides speed command to the eye (converted into torque signals by the neural integrator (NI)). Since the NI functions to convert the speed command into torques, here we will describe the parameters (A and B) of the combined NI and eye dynamics. $A$ and $B$ are two state

matrices (as in [16]):

$$A = \begin{pmatrix} 0 & 1 \\ 0 & -\dfrac{1}{0.005} \end{pmatrix}$$

$$B = \begin{pmatrix} 0 \\ \dfrac{1}{0.005} \end{pmatrix}$$

(2.1)

Note that in [16], the author described the eye dynamics (transformation of torques into eye state 'x(t)') using a second-order differential equation (with time-constants of 145ms and 5ms) while we show Eq 2.1 (transformation between input to the NI 'u(t)' and the eye state 'x(t)'). However, in [16], the second-order eye dynamics was ultimately combined with the dynamics of NI. Such a combined NI and eye dynamics will ultimately attain the same form as our Eq 2.1. We only show the combined NI and eye dynamics with 'u(t)' as input and eye state 'x(t)' as the output for brevity. Overall, this equation transforms the input speed commands into actual eye speed with negligible time-constant owing to the NI compensator that cancels most of the oculomotor inertia.

## Brainstem control

The basic brainstem controller (left part of Fig 1A) without the cerebellum is split in two distinct functional parts: first, a burst generator (BG) receives inputs about visual target displacement and internal feedback from a displacement integrator (DI) and subsequently produces speed commands (referred to as IFL). Second, a NI block that converts the speed commands from the BG into torque signals for the oculomotor plant, so actuating the eye model. Such control loop implementation has been taken from [16]. The BG output activity is defined as:

$$u(t) = a\left(1 - e^{\frac{-(yc+yd-P_{est}(t))}{\sigma}}\right)$$

(3.0)

Where "$u(t)$" is the speed command at time ($t$), "$a$" is the burst amplitude, here set to 1100 Hz (1 Hz corresponds to 1 deg/sec speed command), "$yd$" is the desired displacement, "$yc$" is the cerebellar contribution, and "$P_{est}(t)$" is an imperfect estimated position of the eye at time "$t$" (from the DI in the IFL), "$\sigma$" is a fixed value equal to 16 [16]. The input to the eye model comes from the inverse dynamic controller, which is the sum of "$u(t)$" and a NI to hold the eye in place.

$P_{est}(t)$ is computed a-priori to build a coarse movement plan, considering that no information is available in real-time from the eye during the saccade. $P_{est}(t)$ is defined as:

$$P_{est}(t) = k \int u \, dt$$

(3.1)

Where "$k \int u \, dt$" is the coarse DI which estimates the current position of the eye, the parameter "$k$" corresponds to the level of coarseness ($k = 1$ corresponds to a perfect estimation). Here $k = 0.72$ is set replicating the monkey oculomotor system during cerebellar inactivation (see [16]).

In the absence of cerebellar contribution, the brainstem-based control scheme leads the eye to overshoot the target, as the movement only stops when the Brainstem internal feedback loop (wrongly) indicates that the eye has reached the target. The cerebellar module is used to

**Table 1.  List of abbreviations.**

| Number | Acronym | Full form |
|---|---|---|
| 1. | PC | Purkinje cell |
| 2. | MF | Mossy Fiber |
| 3. | GrC | Granule Cell |
| 4. | MLI | Molecular layer interneuron |
| 5. | LTP | Long term potentiation |
| 6. | LTD | Long term depression |
| 7. | STDP | Spike timing dependent plasticity |
| 8. | PF | Parallel Fiber |
| 9. | SSpike | Simple spike |
| 10. | CSpike | Complex spike |
| 11. | IO | Inferior Olive |
| 12. | CF | Climbing fiber |
| 13 | DCN | Deep cerebellar nuclei |
| 14 | SC | Superior Colliculus |
| 14 | $W_{PF-PC_b}$ | Synaptic weights between parallel fibers and burst Purkinje cells (burst cells are the PC neurons with high excitatory connections from the granule cells and relatively lower molecular layer inhibition) |
| 15 | $W_{PF-PC_p}$ | Synaptic weights between parallel fibers and pause Purkinje cells (pause cells are the PC neurons with low excitatory connections from the granule cells and relatively higher molecular layer inhibition) |

detect the erroneous brainstem control, and subsequently generate corrections to improve movement quality. For an overview of the acronyms in this article, refer to Table 1.

## Cerebellum model

We modified the cerebellar spiking neural network described in [42–44] to adapt it to our experiment. In the right part of Fig 1A, we depicted different components of the simulated cerebellum spiking neural network.

The model contains leaky integrate and fire (LIF) point neurons where connections and delays are tuned to simulate the tight and precise organization and resting-state functionality of the cerebellar microcircuitry and to integrate the cerebellar cortex with deep cerebellar nuclei, including 10 types of neuronal types. In our model we used:

- 89 Mossy fibers (MF)

- 1804 glomeruli (glom)

- 22675 Granule cells (GrC)

- 54 Golgi cells (GoC)

- 68 Purkinje cells (PC)

- 2000 Stellate cells (SC)

- 98 Basket cells (BC)

- 6 Deep cerebellar nuclei (DCN)

- 6 Deep cerebellar nuclei interneurons (DCNint)

- 2 Inferior olive neurons (IO)

This cerebellar network has been reconstructed and validated in various versions using two different neural network simulators: NEST [77] and NEURON [78]. NEST is a simulator for point-neuron models, while NEURON can simulate more complex biophysical neuron models. In our work, we used version-2 of the NEST simulator. All simulations were performed on a shared server mounting a Ryzen 3950x and 64 GB of RAM, using 24 threads. The most computational demanding simulation (showed in Fig 2) including all the plasticities considered in this work took about 1 hour to simulate 345 second, with a constant time resolution of 1 ms, which has been used for all the simulations.

The input related to the target displacement is encoded by the MFs (see Fig 1A), each one associated with a Gaussian receptive field tuned for a specific displacement range (with uniformly separated means in the range of [0, 20] deg, and standard deviation of 5 degree). The input is given to the cerebellum as an external current to MF neurons. The input current strength starts at its maximum, lasting for the entire movement time of the saccades generated by the control loop. In the last 50 ms, the intensity of the input decreases linearly reaching 0 in the last millisecond. Though MF activity can be classified into long-leading and short-leading types as in previous models [18, 22], we omit such distinctions in this study for the sake of simplicity and emphasis on emulating the PC population activity.

MFs form excitatory connection with Granule cells, which in turn excite both Purkinje cells (PCs) and Molecular layer interneurons (MLIs, composed of BC and SC) by means of granule cell axonal extensions known as parallel fibers (PF). In addition to the PF excitation, the PCs receive inhibitory projections from the MLIs. Hence, PCs activity is shaped by two pathways: a direct pathway from excitatory PF connections, and an indirect pathway from inhibitory MLIs. In this work, PCs have been split in two equi-populated groups: burst PCs (with a baseline firing rate of 5 Hz) and pause PCs (baseline firing rate of 200 Hz, obtained by increasing the constant current input). Having a higher baseline firing rate on the pause subpopulations ensures that there is sufficient level of pause capability until the MLI inhibition forces a 0 Hz firing rate through inhibition. The two PC subpopulations are connected to the same neuron's groups, however the weight of the inhibitory connections between the MLI and the pause PCs is set to be 6 times higher than the burst PCs to induce a dominant pause effect.

It is worth noticing that we used only the plasticity between the parallel fibers and the Purkinje cells (PF-PC synapses), while we did not include other plasticity mechanisms (e.g., between MF and DCN or between PC and DCN) that are available in the cerebellar model. In fact, we wanted, to better isolate the role of the PF-PC plasticity. It should be also considered that PF-PC plasticity is implicated in rapid adaptation, while the other plasticity mechanisms are implicated in memory consolidation at slower time scale [65] which is out of the scope of this work.

In the model, PCs inhibit DCN, which are the output of the cerebellum. In our work cerebellar contribution ("*yc*" in Eq 3.0) is determined by the voltage of DCN, while we blocked the DCNs from firing (spiking), by setting the spiking threshold ($V_{th}$) to a very large value. We preferred to use the voltage of the DCN instead of computing DCN average firing rates. This is useful for two reasons. First, we can focus on whether the PC output is sufficient to modulate eye speed, and end displacement without adding complex spiking properties in the DCN circuit. Second, by avoiding spike-to-rate transformation at the DCN level, we increase stability of the simulations and reduce the effort for hyper-parameter tuning. The output is derived by subtracting the DCN basal voltage, (obtained by running the cerebellar simulation, without inputs, for 500 ms) to the current DCN voltage. We do not expect the results to change significantly even if we add spiking property to the DCN's provided the DCN's transmit the PC activity reliably to the brain stem burst neurons in terms of average firing rate. At the end of the eye movement simulation the foveal error is calculated and if the eye overshoots the target, then

the IO neurons generate error driven spikes. The IO forms excitatory connections with the PCs, leading to complex spikes (CSpikes) in PCs. The spike probability of the IO model is related to the foveal error measured after a given saccade, according to the following probability density function:

$$P_{IO_y spike}(e) \begin{cases} 0, & if \ e < 0 \\ 0.2e, & if \ 0 \le e \le 1 \\ 0.2, & if \ e > 1 \end{cases} \tag{4.0}$$

Where "$e$" is the magnitude of foveal error in the latest trial, $P_{IO_y spike}(e)$ is the probability of spike of $IO_y$, where "$y$" denotes the spatial error direction for which the given IO exhibits maximal firing rate. Such a notation is more important for modeling vectorial movements in both horizontal and vertical directions where multiple IO's can be activated at different levels based on the direction of error. Essentially, Eq 4.0 generates '0' probability spikes at IO for accurate movements, a low firing rate proportional to the foveal error for low errors (between 0 and 1 degree), and a constant but still low firing rate for errors above 1 degree. This is qualitatively similar to the probability of complex spikes observed experimentally in the PCs following foveal errors [33] and is visualized in Fig 1B

## Simulation paradigm

Each trial consists of 3 parts (S3 Fig): Inter-trial Period, Movement period, and Post movement period. During the inter-trial period, lasting 300 ms, there is no MF nor IO activity. The movement period contains the anticipatory period where the cerebellum receives the target displacement information, and the actual movement of the eye. The anticipatory activity is caused by the target information given to the MFs ~30ms before the start of the movement. The Post movement period is composed of a resting time with no inputs to MFs and IO (lasting ~100 ms to account for foveal/visual error processing delays), followed by a spiking period where IOs might produce spikes depending on whether there is a perceived foveal error (lasting ~50 ms). The delay from the end of movement period to the activity of the IO allows for PF spikes to be compared to the time of occurrence of IO spikes in the post-movement period in order to produce LTD in the respective PF synapses projecting onto the PCs. The difference between the PF and IO spiking determines how much LTD is caused in the PF-PC synapses (see kernel function in the learning rule Eq 5.7). For a more detailed explanation of the learning rule please refer to [79].

## Plasticity rule

The synaptic weights between the parallel fibers and the Purkinje cells evolve in time, as described by the following equations:

$$W_{PF_iPC_j}(t) = W_{PF_iPC_j}(t-1) + \Delta W_{PF_iPC_j}(t) \tag{5.0}$$

Where $W_{PF_iPC_j}(t)$ is the weight between the parallel fiber $i$ ($PF_i$) and the purkinje cell $j$ ($PC_j$) at time $t$, $\Delta W_{PF_iPC_j}(t)$ is the amount of weight change at time $t$. $\Delta W_{PF_iPC_j}(t)$ is determined by dual plasticity rules, given by:

$$\Delta W_{PF_iPC_j}(t) = LTP_{PF_iPC_j}(t) + LTD_{PF_iPC_j}(t) \tag{5.1}$$

Where $LTP_{PF_iPC_j}(t)$ and $LTD_{PF_iPC_j}(t)$ are the long-term potentiation and long-term depression spike timing-based plasticity mechanisms respectively.

The LTP part is defined as:

$$LTP_{PF_i PC_j}(t) = \alpha_{PC_j} \delta_{PF_i}(t) \tag{5.2}$$

Where $\alpha_{PC_j}$ is the learning rate for the LTP, associated to the $PC_j$. Notably, $\alpha_{PC_j}$ is different for PC pause and burst (we refer to Eq 6.0 for a more detailed explanation). $\delta_{PF_i}(t)$ is a term which takes into consideration the activity of the $PF_i$:

$$\delta_{unit_z}(t) = \begin{cases} 1, if\ unit_z\ is\ active \\ 0, else \end{cases} \tag{5.3}$$

A unit is considered active at time $t$ if emits an action potential at that time.

The LTD part is defined as:

$$LTD_{PF_i PC_j}(t) = \delta_{IO_y}(t)\beta_{PC_j} \int_{-\infty}^{t_{iopre(t)}} K(t-x)\delta_{PF_i}(t-x)\ dx \tag{5.4}$$

Where $\beta_{PC_j}$ is the learning rate for the LTD, associated to the $PC_j$. Notably, it is different for PC pause and burst (we refer to Eq 6.0 for a more complete explanation). $\delta_{IO_y}(t)$ considers the activation of the Inferior Olive $y$ ($IO_y$) connected to the $PC_j$, $t_{iopre(t)}$ is the time of the last spike of $IO_y$:

$$t_{iopre(t)} = \begin{cases} t_{iopre(t-1)}, if\ \delta_{IO_y}(t) = 0 \\ t, else \end{cases} \tag{5.5}$$

$K(t-x)$ is a kernel function related to the time distance between the spike time of $PF_i$ and the one of $IO_y$, defined as:

$$K(s) = \begin{cases} Kint(s),\ if\ s \geq 0\ and\ s \leq 200 \\ 0,\ else \end{cases} \tag{5.6}$$

Where $Kint(s)$ [42, 80] is equal to:

$$Kint(s) = e^{-\frac{s-150}{1000}} * \frac{sin\left(2 * \pi * \frac{(s-150)}{1000}\right)^{20}}{1.2848} \tag{5.7}$$

The kernel function ensures that the PF connections that produce spikes which lead the IO spikes by 150 ms incur largest LTD, while the amount of LTD rapidly decreases as the time difference changes. We set the peak of the kernel function at 150 ms because this value has been previously used to model different eye behaviors such as the VOR and eye-blink conditioning [41]. However, we emphasize that this parameter can be adjusted based on future experimental evidence specifically from saccade behaviors.

## Relationship between the model cerebellar activity and movement control signals

The cerebellum in the model indirectly influences the motor/speed drive ($\Delta u_n[t]$) (at time $t$ and trial number $n$) through two distinct PC subpopulations (burst subpopulation and pause subpopulation), that project onto a common DCN and receive common connections from the IO. Importantly, we constrain our PC populations to generate a horizontal speed drive to the brainstem circuit, while receiving horizontal foveal error information from its CF afferents. Such a simplification is necessary to focus on how movements in a specific direction and visual

errors in specific directions can be improved by adjusting firing rate of local PC populations. In reality, each such local PC populations can provide motor drive in several directions (as a cosine function for equidistant, and radially placed targets) and can respond to errors occurring in several directions (as a gaussian function with peak firing rate at the preferred error direction) [45]. However, including such directional motor control increases the model complexity, and does not provide additional results for our hypothesis that dual plasticity processes in PF-PC synapses are sufficient to decrease the end foveal error and increase the peak speed of saccades. We show in the results that the model emulates experimental recordings of PC populations, despite the imposed constraints.

Bearing the above-mentioned constraints in mind, the relationship between the burst and pause PC population activities (*b* and *p* respectively) and the total speed drive to the oculomotor system can be written as:

$$u_n[t] = f(b_n[t - \varepsilon] + p_n[t - \varepsilon]) \tag{5.8}$$

Where *f* represents downstream transformation of the PC population response through the DCN and the brainstem circuit. *t* is the time instant and $\varepsilon$ is the time-lag between the PC activity and the actual motor drive (set to 30 ms). The burst and pause PC responses, at a given time instant are dependent on the granule cell activity transmitted by the parallel fibers (PF), given by:

$$b_n(t) = \sum_b W_{PF-PC_b}(n) * PF_b(t) \tag{5.9}$$

$$p_n(t) = \sum_p W_{PF-PC_p}(n) * PF_p(t) \tag{6.0}$$

Where $W_{PF-PC_b}(n)$ represents the connection strengths of the PF-PC synapses onto the burst PC populations and $W_{PF-PC_p}(n)$ represents the connection strengths of the PF-PC synapses onto the pause PC populations. *PF(t)* represents the firing rate of PFs projecting onto the PC populations either through direct excitatory projections or through indirect inhibitory projections from molecular interneurons (MLIs) (see Fig 1).

By adjusting the PF-PC weights $W_{PF-PC_b}$ and $W_{PF-PC_p}$, it is possible to modulate the burst and pause PC population firing rates, which will influence the motor drive $u_n[t]$ to the eye through downstream DCN and brainstem circuits. The changes in burst and pause PC firing rates from *n-1th* trial to *nth* trial can be represented as

$$b_n(t) - b_{n-1}(t) \propto W_{PF-PC_b}(n) - W_{PF-PC_b}(n - 1) \tag{6.1}$$

$$p_n(t) - p_{n-1}(t) \propto W_{PF-PC_p}(n) - W_{PF-PC_p}(n - 1) \tag{6.2}$$

From Eqs 5.0–5.4, we know that the trial-by-trial change in weights can be regulated by the combined LTP and LTD plasticity mechanisms.

Recent experiments indicate that the IO driven late-CSpike responses occurring after the end of the eye movement are correlated with the end/late foveal error [33]. It is reasonable to relate the late-CSpike responses to be arising from the IO activity in our model, which according to Eqs 5.4–5.7 produces an LTD in the direct PF-PC synapses when a positive error occurs. For simplicity, positive error can be assumed to arise during overshooting movement that requires pulling the eye position inwards to correct the gaze direction. Note that in the

biological cerebellum, positive/negative errors are preferentially encoded in the CSpike activity of distinct PC populations [33].

## Tuning the learning rate parameters of burst and pause PC subpopulations

The amount of change in synaptic weights is dependent on both LTP and LTD in PF-PC synapses (Eq 5.1). If either of the LTP or LTD dominate the plasticity process due to disproportionately larger learning rates, then the movements resulting from such skewed plasticity processes will diverge from the optimal scenario where we expect increase in peak speed and decrease in end foveal error. Notably, not all parameters for the LTP and LTD lead to increase in both peak speed and decrease in end foveal error, as one term may dominate the other for a large range of LTP, LTD parameters. Hence it is crucial to appropriately tune the learning rate parameters $\alpha$ and $\beta$ of LTP and LTD respectively for generating optimal movements. Especially the learning rate parameters are not necessarily the same for burst ($\alpha_{PC_b}$, $\beta_{PC_b}$) and pause ($\alpha_{PC_p}$, $\beta_{PC_p}$) PC subpopulations. We explored for a range of parameter values ($\alpha_{PC_b}$, $\beta_{PC_b}$, $\alpha_{PC_p}$, $\beta_{PC_p}$). Briefly in the PF-PC excitatory synapses, when the movement errors are high in magnitude then the effect of error-independent LTP on the weight update needs to smaller when compared to that of error-dependent LTD such that such that larger reductions in movement errors occur due to a larger LTD component. Furthermore, the LTP dominates to increase the peak speed component when the errors are reduced below a certain threshold (we considered the threshold to be e = 0.5 deg, see Fig 1B).

We found a range of ($\alpha_{PC_b}$, $\beta_{PC_b}$, $\alpha_{PC_p}$, $\beta_{PC_p}$) parameters that led to increase in peak speed and decrease in foveal error through a guided binary search. In the first step, we set only two parameters values, LTP and LTD (or $\alpha_{PC}$ and $\beta_{PC}$), commonly across burst and pause PC subpopulations. These parameters were selected from multiple solutions of LTP and LTD values found in [81] on different sensorimotor tasks. Note that these initial parameters were not fine-tuned for the saccade task, but were well tested in other sensorimotor tasks [81]. Notably, even without fine tuning, a large range of parameter values were observed to modulate atleast the foveal error. Subsequently, for finding the optimal parameter tuning that influence both movement speed and error, we performed a search in the vicinity of the initial parameters (testing the parameter changes in the range of [$10^{-9}$–$10^{-4}$] for $\alpha_{PC_b}$ and $\alpha_{PC_p}$, and [$10^{-9}$–$10^{-4}$] for $\beta_{PC_b}$, $\beta_{PC_p}$). Note that while changing LTP parameter, the LTD parameter was held constant, and vice-versa so that a set of parameters that can influence error and speed was found. We followed two simple steps to evaluate the quality of parameter selection. First, we visually verified if the selected parameters produced information flow from MF input activation to the PC output. i.e., we verified that the norm of PC population activity was different from the baseline and the SSpikes persisted at least until as long as the MF activity persisted. Second, we evaluated the parameters by running the saccades to 10-degree target for multiple trials, and considered the parameters as a good set if the foveal error after convergence of training was ~0.5 degree, and if the peak speed was higher at the end of 150–200 trials was higher than the first 10 trials. We manually selected a subset among the parameters found by binary search and used them for simulations (Table 2). In Fig 1B, we visualized the distinct effect of the learning

**Table 2. Plasticity parameters of LTP and LTD for different PC subpopulations.** Note that these parameters are not the sole measure of the amount of learning per unit error, but are in fact the constants used to scale and adjust the direction of PF-PC synaptic weight update.

|  | LTD | LTP |
|---|---|---|
| **Pause PC** | - 6.4 • $10^{-6}$ | 5.2 • $10^{-7}$ |
| **Burst PC** | - 8.1 • $10^{-7}$ | 3.915 • $10^{-6}$ |

rule with exemplary learning rate parameters on the activity of burst and pause PC subpopulations. The change in PF-PC weights ($\Delta W_{PF_i PC_j}$) of a given PC subpopulation depended on the ratio between the two learning rates ($\alpha_{PC_j}$, $\beta_{PC_j}$) and on the foveal error occurring after each saccade.

As a result of the tuned learning rates in the simulations, the sign of $\Delta W_{PF_i PC_j}(t)$ for PC pause and PC burst were concordant if the error was equal to 0.0 or higher than 1.0. Conversely, the weight changes, $\Delta W_{PF_i PC_j}(t)$, in pause and burst subpopulations were discordant if the error was around 0.5 deg. In the discordant zone PF-PC connections of pause subpopulations decreased their excitatory weights due to a dominant LTD process (increasing the amount of drop in firing rate below baseline), while PF-PC connections of the burst subpopulations increased their excitatory weights due to a dominant LTP process (increasing their firing rate above baseline).

The mechanistic description of long-term plasticity changes that we have used is relatively simple and as previously used in several protocols where the cerebellum is crucially involved in generating motor adaptation [41, 42, 65, 81, 82]. However, it is difficult to map the learning rate parameters to specific biological components. For instance, in glutamatergic synapses, LTP and LTD are produced with a change (increase and decrease, respectively) in the number of AMPA receptors present in the post-synaptic density [83]. These changes are regulated by complex intracellular chain reactions involving multiple proteins. Each step of these processes has different timings and dynamics which are, in the end, responsible of the precise and differential time course of LTP and LTD.

## Supporting information

**S1 Fig. PC population activity (SSpike firing rate) across training trials (y-axis) and across each trial duration (x-axis) t = 0ms corresponds to saccadic movement onset.** The heatmap corresponds to the mean firing rate of PC subpopulations: a) burst PCs, b) pause PCs, c) all PCs.
(TIF)

**S2 Fig. Change in movement properties to multiple targets across training trials in dual plasticity and only-LTD conditions.**
(TIF)

**S3 Fig. Schematic of each trial.** Each trial is split in 3 parts: Inter-trial period (left), movement period (center) and the post movement period (right). In each of these parts the eye plant activity (top) and the spiking cerebellum activity (bottom) are presented. During the inter-trial period there is no activity in the eye model, while the spiking cerebellum model does not receive any inputs (baseline spiking is still present). In the movement period the cerebellum receives the target displacement information through MFs and the eye activity is split in 2 parts: an anticipatory period and a movement period. The post movement period does not have eye model activity, while the cerebellum model has a period of no activation and one in which, depending on the foveal error, the IO has a certain probability of spiking. White and purple rectangles correspond to absence or presence of activation, respectively.
(TIF)

## Author Contributions

**Conceptualization:** Lorenzo Fruzzetti, Hari Teja Kalidindi, Claudia Casellato, Egidio Falotico, Egidio D'Angelo.

**Data curation:** Lorenzo Fruzzetti, Hari Teja Kalidindi, Alberto Antonietti.

**Formal analysis:** Lorenzo Fruzzetti, Hari Teja Kalidindi.

**Funding acquisition:** Egidio Falotico, Egidio D'Angelo.

**Investigation:** Lorenzo Fruzzetti, Hari Teja Kalidindi.

**Methodology:** Lorenzo Fruzzetti, Hari Teja Kalidindi, Alberto Antonietti, Cristiano Alessandro, Alice Geminiani, Claudia Casellato, Egidio Falotico, Egidio D'Angelo.

**Resources:** Claudia Casellato, Egidio Falotico, Egidio D'Angelo.

**Software:** Lorenzo Fruzzetti, Hari Teja Kalidindi, Alberto Antonietti, Alice Geminiani, Claudia Casellato.

**Supervision:** Claudia Casellato, Egidio Falotico, Egidio D'Angelo.

**Validation:** Lorenzo Fruzzetti, Hari Teja Kalidindi, Alberto Antonietti, Cristiano Alessandro, Alice Geminiani, Claudia Casellato, Egidio Falotico, Egidio D'Angelo.

**Visualization:** Lorenzo Fruzzetti, Hari Teja Kalidindi.

**Writing – original draft:** Lorenzo Fruzzetti, Hari Teja Kalidindi, Alberto Antonietti, Claudia Casellato, Egidio Falotico, Egidio D'Angelo.

**Writing – review & editing:** Lorenzo Fruzzetti, Hari Teja Kalidindi, Alberto Antonietti, Cristiano Alessandro, Alice Geminiani, Claudia Casellato, Egidio Falotico, Egidio D'Angelo.

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
