## [Decision Letter · Decision Letter 0]

1 Jun 2022

Dear Mr. kalidindi,

Thank you very much for submitting your manuscript "Dual STDP processes at Purkinje cells contribute to distinct improvements in accuracy and vigor of saccadic eye movements." for consideration at PLOS Computational Biology.

As with all papers reviewed by the journal, your manuscript was reviewed by members of the editorial board and by several independent reviewers. In light of the reviews (below this email), we would like to invite the resubmission of a significantly-revised version that takes into account the reviewers' comments.

While all reviewers were generally positive, they had some valid concerns that the authors need to address. Specifically, reviewers thought that more details regarding the learning process and how model simulations match saccade data would be important. I would encourage authors to consider reviewers 2 and 3 in particular, as they have raised some critical concerns surrounding the usefulness and appropriateness of the model.

We cannot make any decision about publication until we have seen the revised manuscript and your response to the reviewers' comments. Your revised manuscript is also likely to be sent to reviewers for further evaluation.

Sincerely,

Gunnar Blohm, Ph.D.

Associate Editor

PLOS Computational Biology

Thomas Serre

Deputy Editor

PLOS Computational Biology

Reviewer's Responses to Questions

**Comments to the Authors:**

Reviewer #1: The study “Dual STDP processes at Purkinje cells …” [PCOMPBIOL-D-22_00441] describes a computational model of the cerebellum that accounts for how dualing plasticity processes contribute to saccade endpoint control and kinematics. They posit 2 learning rules – LTD which is error-driven, and LTP which is error-independent – as well as upward and downward modulated Purkinje cells. This cerebellar module receives target related information and sends output, via the DCN to the saccade burst generator networks in the brainstem, to influence saccadic eye movements. The upshot of their findings is that this arrangement facilitates movement optimization, enhancing endpoint control and speeds up the movements. Exploration of the substrates of this improved motor behavior showed that LTP and LTD mediate different aspects of motor enhancement in their model and they require integration between upward and downward going Purkinje cells. I thoroughly enjoyed this manuscript, found the results intriguing, and think the study will be of interest in the field of cerebellar learning and control systems theory. Below I outline a few suggestions that may improve the manuscript.

1. Line 301-307. The authors describe the effects on Purkinje cells of turning on and off LTP and LTD. It is not clear why turning off LTP makes PC pausers increase firing, though later the opposite is not seen when LTD is turned off. It would be helpful to provide an intuition for the reader here.

2. A little more information about the learning rules should be presented in the main text.

3. The model’s DCN output, though essential, is weakly described. I did not follow what the tuning of its output(s) are, how they influence the burst generator, nor their integration rules of Purkinje cells. I did not follow how much mixing of PC types there was in each DCN neuron. I did not follow how the speed and endpoint control are mediated by the same DCN neurons (or if so). If the PCs exert vectorial learning, do the DCN neurons exert vectorial control? If so, what are the assumptions about their directional tuning? Can any predictions be made from this model on how DCN neurons fire to control movement?/what their tuning might reflect? Etc? Finally, does model include the DCN output neurons inhibitory onto the IO, as are known to exist? If not, it would be worth discussing limitations of the model without this feedback circuit element.

4. In the section 253-268 (approximately), on predictive control, it was hard to appreciate why reducing MF activity was used to test this. Better exposition setting up this experimental design is warranted.

Reviewer #2: This study investigates the neural mechanism underlying control of saccade vigor and accuracy. They have developed a novel model of cerebellar control employing spike-timing dependent plasticity rules and shown via simulations that a control policy using a dual plasticity mechanism predicts changes in saccade accuracy and vigor. Specifically, they show that, with training, the model reduces movement error to a 10 degree target, while also increasing saccade peak velocity. They then show that these effects rely differentially on LTD (for accuracy) and LTP (velocity increase).

Generally, it is interesting to see that the simulation predicts these independent effects of LTD and LTP on accuracy and peak velocity, respectively. However, my main concern is that I am not convinced that the behavior they see in simulation has been observed experimentally. In their model, initial saccades to a 10 degree target are slow and exhibit overshoot. With training, over many trials, overshoot errors are reduced and saccades increase in peak velocity. While there are many experimental demonstrations of error reduction with training, is there evidence of a concomitant increase in peak velocity? The paper does not clarify whether or not this is the case or any reason why we should expect to see an increase in peak velocity with lower amplitude movements. Additionally, there is no clear rationale presented for why the cerebellum is involved in the setting and control of saccade velocity. In the points below I explain the many reasons I am confused by their claims and ultimately left unconvinced by the validity of the model.

1. The model predicts a reduction in overshoot and increase in peak velocity with training. There is no rationale presented for why we should expect the latter. (a) What is the rationale for why peak velocity increases with training? Importantly, amplitude is also reduced with training. Peak velocity increases despite reductions in saccade amplitude. (b) Again, the rationale here is not clear to me. Are the authors suggesting that peak velocity is modulated independently of the main sequence, and will increase despite reductions in amplitude? (c) The authors describe that these velocity increases demonstrated that the movement is ‘optimized’ or ‘improved’, but they do not explain what the cost function or metric of improvement is. Overall, it would have been helpful to initially lay out the expected model behavior regarding error and peak velocity and the underlying rationale/evidence.

2. The study suggests that the cerebellum plays a role in planning and dictating saccade peak velocity. This is not in contrast to current evidence suggesting that the cerebellum implements a plan determined by the superior colliculus. It would be helpful for the authors to clarify and expand upon the rationale for cerebellar control of saccade peak velocity.

3. A suggestion to help clarify hypotheses and validate the model would to demonstrate that the model can simulate the main sequence. If this has been shown in an earlier paper, then a statement as such would suffice.

4. A lesser point is the use of the terms ‘vigor’ and ‘accuracy’.

a. Vigor is usually described as peak velocity normalized to movement amplitude. This allows one to distinguish between change in peak velocity due to reward from the well-established change in peak velocity due to movement amplitude. The authors use the terms interchangeably here, but I recommend using only peak velocity, unless they are normalizing to the movement amplitude.

b. Accuracy is often used to describe the variability of movement endpoints over may trials and thought of in the context of speed-accuracy trade-offs in movement. I understand that the single trial overshoot described in this paper is also a reflection of accuracy, but it more akin to the error observed in a movement adaptation study than a speed accuracy study. Thus to improve the clarity, it would be helpful to explain that error, overshoot, and accuracy are all the same here and to define what is meant by them.

Reviewer #3: Fruzzetti et al. proposes a biologically realistic model of the cerebellar computational dynamics. Importantly, the manuscript demonstrates the potential mechanisms by which the cerebellum can leverage heterogeneous spike-timing-dependent plasticity (STDPs) to optimize oculomotor adaptation.

The authors have done a great job setting up the research question and the motivations for developing the proposed model. I also especially find the introduction and discussion to be interesting and informative. The primary questions I have regarding the manuscript concern the modelling choices adopted and the statistical analyses performed. Below, I have included specific concerns that I hope the authors could help address.

1. I believe the proposed modelling schemes and the learning rules are intriguing and innovative. However, it would be helpful for the authors to consider adding citations and more justifications for specific modelling/parameter choices implemented in the proposed model. For example, the authors chose to use the voltage of the deep cerebellar nuclei (rather than the average firing rates) to determine the cerebellar contribution (yc). Could you please provide additional information regarding this modeling choice? Additionally, could the authors discuss how this affects the overall observed model dynamics? Additionally, how would the model dynamics and conceptual interpretation of the observed dynamics change if the average firing rates were used instead?

2. Regarding the plasticity rule, are there competing theories for what the plasticity rule should be for a cerebellar system? It will be helpful for readers to understand the limitations of the model dynamics if the relevant context of other possible plasticity rules is provided.

3. Information regarding statistical analyses appears to be missing. What kinds of statistical tests were performed to evaluate the model dynamics (e.g., the behavioral and neural effect of cerebellar learning by the PF-PC plasticity mechanisms) and what were the results? Such information is important for readers to understand different aspects of the models and their emergent behaviors. For example, it is unclear if the cerebellar spiking neural network learning (as induced by the dual plasticity) led to significant improvement in accuracy and vigor across multiple movements over time.

4. The authors discussed tuning of the learning rate parameters of LEP and LTD as an essential step towards generating optimal behavior, especially in the context of both burst and pause subpopulations. However, it is unclear how such tuning was performed in the present manuscript. Please consider including additional information regarding the different steps taken to ensure optimal tuning. Additionally, could the authors discuss the mechanisms by which this tuning process is achieved in nature?

5. What are the technical limitations of the proposed model that the readers should be aware of/take into account as they interpret the observed results?

**Have the authors made all data and (if applicable) computational code underlying the findings in their manuscript fully available?**

Reviewer #1: Yes

Reviewer #2: **No: **The link provided is not active

Reviewer #3: **No: **It is unclear if the authors made the data/code available.

PLOS authors have the option to publish the peer review history of their article (what does this mean?). If published, this will include your full peer review and any attached files.

Reviewer #1: No

Reviewer #2: No

Reviewer #3: No
---

## [Decision Letter · Decision Letter 1]

13 Sep 2022

Dear Mr. kalidindi,

We are pleased to inform you that your manuscript 'Dual STDP processes at Purkinje cells contribute to distinct improvements in accuracy and speed of saccadic eye movements.' has been provisionally accepted for publication in PLOS Computational Biology.

Best regards,

Gunnar Blohm, Ph.D.

Academic Editor

PLOS Computational Biology

Thomas Serre

Section Editor

PLOS Computational Biology

Reviewer's Responses to Questions

**Comments to the Authors:**

Reviewer #1: The authors have thoroughly and satisfactorily addressed my concerns. I think this study provides a solid conceptual advance in how bursting and pausing Purkinje neurons work together with competing plasticity rules to enhance motor behavior.

Reviewer #2: The authors have responded thoughtfully and in detail to all of my concerns. I hank the authors for their efforts and have no further concerns.

Reviewer #3: The authors have thoroughly addressed all of my concerns. Given the complexity of different comparisons being studied, my only other minor suggestion is for the authors to please include the type of statistical tests used (e.g., paired t-test instead of t-test) as well as corresponding test values and/or degree of freedom when necessary.

**Have the authors made all data and (if applicable) computational code underlying the findings in their manuscript fully available?**

Reviewer #1: Yes

Reviewer #2: Yes

Reviewer #3: Yes

PLOS authors have the option to publish the peer review history of their article (what does this mean?). If published, this will include your full peer review and any attached files.

Reviewer #1: No

Reviewer #2: No

Reviewer #3: No

---

## [Editor Report · Acceptance letter]

28 Sep 2022

PCOMPBIOL-D-22-00441R1 

Dual STDP processes at Purkinje cells contribute to distinct improvements in accuracy and speed of saccadic eye movements.

Dear Dr kalidindi,

I am pleased to inform you that your manuscript has been formally accepted for publication in PLOS Computational Biology. Your manuscript is now with our production department and you will be notified of the publication date in due course.

With kind regards,

Zsanett Szabo
